# Provable Guarantees for Self-Supervised Deep Learning with Spectral Contrastive Loss

**Jeff Z. HaoChen**
Stanford University
jhaochen@stanford.edu

**Colin Wei**
Stanford University
colinwei@stanford.edu

**Adrien Gaidon**
Toyota Research Institute
adrien.gaidon@tri.global

**Tengyu Ma**
Stanford University
tengyuma@stanford.edu

## Abstract

Recent works in self-supervised learning have advanced the state-of-the-art by relying on the *contrastive learning* paradigm, which learns representations by pushing positive pairs, or similar examples from the same class, closer together while keeping negative pairs far apart. Despite the empirical successes, theoretical foundations are limited – prior analyses assume conditional independence of the positive pairs given the same class label, but recent empirical applications use heavily correlated positive pairs (i.e., data augmentations of the same image). Our work analyzes contrastive learning without assuming conditional independence of positive pairs using a novel concept of the *augmentation graph* on data. Edges in this graph connect augmentations of the same datapoint, and ground-truth classes naturally form connected sub-graphs. We propose a loss that performs spectral decomposition on the population augmentation graph and can be succinctly written as a contrastive learning objective on neural net representations. Minimizing this objective leads to features with provable accuracy guarantees under linear probe evaluation. By standard generalization bounds, these accuracy guarantees also hold when minimizing the training contrastive loss. Empirically, the features learned by our objective can match or outperform several strong baselines on benchmark vision datasets. In all, this work provides the first provable analysis for contrastive learning where guarantees for linear probe evaluation can apply to realistic empirical settings.

## 1 Introduction

Recent empirical breakthroughs have demonstrated the effectiveness of self-supervised learning, which trains representations on unlabeled data with surrogate losses and self-defined supervision signals [4, 6, 10, 14, 23, 24, 35, 38, 41, 42, 50–52]. Self-supervision signals in computer vision are often defined by using data augmentation to produce multiple views of the same image. For example, the recent contrastive learning objectives [3, 12, 13, 15, 22] encourage closer representations for augmentations (views) of the same natural data than for randomly sampled pairs of data.

Despite the empirical successes, there is a limited theoretical understanding of why self-supervised losses learn representations that can be adapted to downstream tasks, for example, using linear heads. Recent mathematical analyses by Arora et al. [3], Lee et al. [28], Tosh et al. [44, 45] provide guarantees under the assumption that two views are somewhat independent conditioned on the label. However, the pair of augmented examples used in practical algorithms usually exhibit a strong correlation, even conditioned on the label. For instance, two augmentations of the same dog image

35th Conference on Neural Information Processing Systems (NeurIPS 2021).

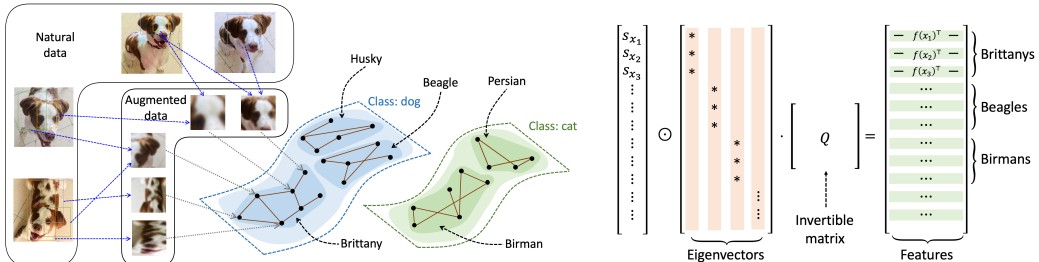

Figure 1: **Left: demonstration of the population augmentation graph.** Two augmented data are connected if they are views of the same natural data. Augmentations of data from different classes in the downstream tasks are assumed to be nearly disconnected, whereas there are more connections within the same class. We allow the existence of disconnected sub-graphs within a class corresponding to potential sub-classes. **Right: decomposition of the learned representations.** The representations (rows in the RHS) learned by minimizing the population spectral contrastive loss can be decomposed as the LHS. The scalar $s_{x_i}$ is positive for every augmented data $x_i$. Columns of the matrix labeled "eigenvectors" are the top eigenvectors of the normalized adjacency matrix of the augmentation graph defined in Section 3.1. The operator $\odot$ multiplies row-wise each $s_{x_i}$ with the $x_i$-th row of the eigenvector matrix. When classes (or sub-classes) are exactly disconnected in the augmentation graph, the eigenvectors are sparse and align with the sub-class structure. The invertible $Q$ matrix does not affect the performance of the rows under the linear probe.

share much more similarity than augmentations of two different random dog images. Thus the existing theory does not explain the practical success of self-supervised learning.

This paper presents a theoretical framework for self-supervised learning without requiring conditional independence. We design a principled, practical loss function for learning neural net representations that resembles state-of-the-art contrastive learning methods. We prove that, under a simple and realistic data assumption, linear classification using representations learned on a polynomial number of unlabeled data samples can recover the ground-truth labels of the data with high accuracy.

The fundamental data property that we leverage is a notion of continuity of the population data within the same class. Though a random pair of examples from the same class can be far apart, the pair is often connected by (many) sequences of examples, where consecutive examples in the sequences are close neighbors within the same class. This property is more salient when the neighborhood of an example includes many different types of augmentations. Prior work [49] empirically demonstrates this type of connectivity property and uses it in the analysis of pseudolabeling algorithms.

More formally, we define the *population augmentation graph*, whose vertices are all the augmented data in the *population* distribution, which can be an exponentially large or infinite set. Two vertices are connected with an edge if they are augmentations of the same natural example. Our main assumption is that for some proper $m \in \mathcal{Z}^+$, the sparsest $m$-partition (Definition 3.4) is large. This intuitively states that we can't split the augmentation graph into too many disconnected sub-graphs by only removing a sparse set of edges. This assumption can be seen as a graph-theoretic version of the continuity assumption on population data. We also assume that there are very few edges across different ground-truth classes (Assumption 3.5). Figure 1 (left) illustrates a realistic scenario where dog and cat are the ground-truth categories, between which edges are very rare. Each breed forms a sub-graph that has sufficient inner connectivity and thus cannot be further partitioned.

Our assumption fundamentally does not require conditional independence and can allow disconnected sub-graphs within a class. The classes in the downstream task can be also somewhat flexible as long as they are disconnected in the augmentation graph. For example, when the augmentation graph consists of $m$ disconnected sub-graphs corresponding to fine-grained classes, our assumptions allow the downstream task to have any $r \le m$ coarse-grained classes containing these fine-grained classes as a sub-partition. Prior work [49] on pseudolabeling algorithms essentially requires an exact alignment between sub-graphs and downstream classes (i.e., $r = m$). They face this limitation because their analysis requires fitting discrete pseudolabels on the unlabeled data. We avoid this difficulty because we consider directly learning continuous representations on the unlabeled data.

We apply spectral decomposition—a classical approach for graph partitioning, also known as spectral clustering [37, 39] in machine learning—to the adjacency matrix defined on the population augmentation graph. We form a matrix where the top-$k$ eigenvectors are the columns and interpret each row of the matrix as the representation (in $\mathbb{R}^k$) of an example. Somewhat surprisingly, we show that this feature extractor can be also recovered (up to some linear transformation) by minimizing the following population objective which is similar to the standard contrastive loss (Section 3.2):

$$\mathcal{L}(f) = -2 \cdot \mathbb{E}_{x,x^+} \left[ f(x)^\top f(x^+) \right] + \mathbb{E}_{x,x'} \left[ \left( f(x)^\top f(x') \right)^2 \right],$$

where $(x, x^+)$ is a pair of augmentations of the same data, $(x, x')$ is a pair of independently random augmented data, and $f$ is a parameterized function from augmented data to $\mathbb{R}^k$. Figure 1 (right) illustrates the relationship between the eigenvector matrix and the learned representations. We call this loss the *population spectral contrastive loss*.

We analyze the linear classification performance of the representations learned by minimizing the population spectral contrastive loss. Our main result (Theorem 3.7) shows that when the representation dimension exceeds the maximum number of disconnected sub-graphs, linear classification with learned representations is guaranteed to have a small error. Our theorem reveals a trend that a larger representation dimension is needed when there are a larger number of disconnected sub-graphs. Our analysis relies on novel techniques tailored to linear probe performance, which have not been studied in the spectral graph theory community to the best of our knowledge.

The spectral contrastive loss also works on empirical data. Since our approach optimizes parametric loss functions, guarantees involving the population loss can be converted to finite sample results using off-the-shelf generalization bounds. The sample complexity is polynomial in the Rademacher complexity of the model family and other relevant parameters (Theorem 4.1 and Theorem 4.2).

In summary, our main theoretical contributions are: 1) we propose a simple contrastive loss motivated by spectral decomposition of the population data graph, 2) under simple and realistic assumptions, we provide downstream classification guarantees for the representation learned by minimizing this loss on population data, and 3) our analysis is easily applicable to deep networks with polynomial unlabeled samples via off-the-shelf generalization bounds.

In addition, we implement and test the proposed spectral contrastive loss on standard vision benchmark datasets. We demonstrate that the features learned by our algorithm can match or outperform several strong baselines [12, 14, 15, 21] when evaluated using a linear probe.

## 2  Additional related works

**Empirical works on self-supervised learning.** Self-supervised learning algorithms have been shown to successfully learn representations that benefit downstream tasks [4, 6, 10, 12, 13, 15, 22–24, 35, 38, 41, 42, 50–52]. Many recent self-supervised learning algorithms learn features with siamese networks [8], where two neural networks of shared weights are applied to pairs of augmented data. Introducing asymmetry to siamese networks either with a momentum encoder like BYOL [21] or by stopping gradient propagation for one branch of the siamese network like SimSiam [14] has been shown to effectively avoid collapsing. Contrastive methods [12, 15, 22] minimize the InfoNCE loss [38], where two views of the same data are attracted while views from different data are repulsed.

**Theoretical works on self-supervised learning.** In addition to works [3, 28, 44, 45] discussed in the introduction, several other works [5, 43, 47, 48] also theoretically study self-supervised learning. The work Tsai et al. [47] prove that self-supervised learning methods can extract task-relevant information and discard task-irrelevant information, but lacks guarantees for solving downstream tasks efficiently with simple (e.g., linear) models. Tian et al. [43] study why non-contrastive self-supervised learning methods can avoid feature collapse. Cai et al. [9] analyze domain adaptation algorithms for subpopulation shift with a similar expansion condition as [49] while also allowing disconnected parts within each class, but require access to ground-truth labels during training. In contrast, our algorithm doesn't need labels during pre-training.

# 3 Spectral contrastive learning on population data

In this section, we introduce our theoretical framework, the spectral contrastive loss, and the main analysis of the performance of the representations learned on population data.

We use $\overline{\mathcal{X}}$ to denote the set of all natural data (raw inputs without augmentation). We assume that each $\bar{x} \in \overline{\mathcal{X}}$ belongs to one of $r$ classes, and let $y : \overline{\mathcal{X}} \to [r]$ denote the ground-truth (deterministic) labeling function. Let $\mathcal{P}_{\overline{\mathcal{X}}}$ be the population distribution over $\overline{\mathcal{X}}$ from which we draw training data and test our final performance. For the ease of exposition, we assume $\overline{\mathcal{X}}$ to be a finite but exponentially large set (e.g., all real vectors in $\mathbb{R}^d$ with bounded precision).[1]

We next formulate data augmentations. Given a natural data sample $\bar{x} \in \overline{\mathcal{X}}$, we use $\mathcal{A}(\cdot|\bar{x})$ to denote the distribution of its augmentations. For instance, when $\bar{x}$ represents an image, $\mathcal{A}(\cdot|\bar{x})$ can be the distribution of common augmentations [12] that includes Gaussian blur, color distortion and random cropping. We use $\mathcal{X}$ to denote the set of all augmented data, which is the union of supports of all $\mathcal{A}(\cdot|\bar{x})$ for $\bar{x} \in \overline{\mathcal{X}}$. As with $\overline{\mathcal{X}}$, we also assume that $\mathcal{X}$ is a finite but exponentially large set, and denote $N = |\mathcal{X}|$.

We will learn an embedding function $f : \mathcal{X} \to \mathbb{R}^k$, and then evaluate its quality by the minimum error achieved with a linear probe. Concretely, a linear classifier has weights $B \in \mathbb{R}^{k \times r}$ and predicts $g_{f,B}(x) = \arg\max_{i \in [r]} (f(x)^\top B)_i$ for an augmented data $x$ ($\arg\max$ breaks tie arbitrarily). Then, given a natural data sample $\bar{x}$, we ensemble the predictions on augmented data and predict:

$$\bar{g}_{f,B}(\bar{x}) := \arg\max_{i \in [r]} \Pr_{x \sim \mathcal{A}(\cdot|\bar{x})} [g_{f,B}(x) = i].$$

Define the *linear probe* error as the error of the best possible linear classifier on the representations:

$$\mathcal{E}(f) := \min_{B \in \mathbb{R}^{k \times r}} \Pr_{\bar{x} \sim \mathcal{P}_{\overline{\mathcal{X}}}} [y(\bar{x}) \neq \bar{g}_{f,B}(\bar{x})] \tag{1}$$

## 3.1 Augmentation graph and spectral decomposition

Our approach is based on the central concept of **population augmentation graph**, denoted by $G(\mathcal{X}, w)$, where the vertex set is all augmentation data $\mathcal{X}$ and $w$ denotes the edge weights defined below. For any two augmented data $x, x' \in \mathcal{X}$, define the weight $w_{xx'}$ as the marginal probability of generating the pair $x$ and $x'$ from a random natural data $\bar{x} \sim \mathcal{P}_{\overline{\mathcal{X}}}$:

$$w_{xx'} := \mathbb{E}_{\bar{x} \sim \mathcal{P}_{\overline{\mathcal{X}}}} [\mathcal{A}(x|\bar{x})\mathcal{A}(x'|\bar{x})] \tag{2}$$

Therefore, the weights sum to 1 because the total probability mass is 1: $\sum_{x,x' \in \mathcal{X}} w_{xx'} = 1$. The relative magnitude intuitively captures the closeness between $x$ and $x'$ with respect to the augmentation transformation. For most of the unrelated $x$ and $x'$, the value $w_{xx'}$ will be significantly smaller than the average value. For example, when $x$ and $x'$ are random croppings of a cat and a dog respectively, $w_{xx'}$ will be essentially zero because no natural data can be augmented into both $x$ and $x'$. On the other hand, when $x$ and $x'$ are very close in $\ell_2$-distance or very close in $\ell_2$-distance up to color distortion, $w_{xx'}$ is nonzero because they may be augmentations of the same image with Gaussian blur and color distortion. We say that $x$ and $x'$ are connected with an edge if $w_{xx'} > 0$. See Figure 1 (left) for more illustrations.

Given the structure of the population augmentation graph, we apply spectral decomposition to the population graph to construct principled embeddings. The eigenvalue problems are closely related to graph partitioning as shown in spectral graph theory [17] for both worst-case graphs [11, 25, 29, 33] and random graphs [1, 30, 34]. In machine learning, spectral clustering [37, 39] is a classical algorithm that learns embeddings by eigendecomposition on an empirical distance graph and invoking $k$-means on the embeddings.

We will apply eigendecomposition to the *population* augmentation graph (and then later use linear probe for classification). Let $w_x = \sum_{x' \in \mathcal{X}} w_{xx'}$ be the total weights associated to $x$, which is often viewed as an analog of the degree of $x$ in weighted graph. A central object in spectral graph theory is the so-called *normalized adjacency matrix*:

$$\overline{A} := D^{-1/2} A D^{-1/2} \tag{3}$$

---

[1]This allows us to use sums instead of integrals and avoid non-essential nuances related to calculus.

where $A \in \mathbb{R}^{N \times N}$ is adjacency matrix with entires $A_{xx'} = w_{xx'}$ and $D \in \mathbb{R}^{N \times N}$ is a diagonal matrix with $D_{xx} = w_x$.[2]

Standard spectral graph theory approaches produce vertex embeddings as follows. Let $\gamma_1, \gamma_2, \cdots, \gamma_k$ be the $k$ largest eigenvalues of $\overline{A}$, and $v_1, v_2, \cdots, v_k$ be the corresponding unit-norm eigenvectors. Let $F^\star = [v_1, v_2, \cdots, v_k] \in \mathbb{R}^{N \times k}$ be the matrix that collects these eigenvectors in columns, and we refer to it as the eigenvector matrix. Let $u_x^* \in \mathbb{R}^k$ be the $x$-th row of the matrix $F^\star$. It turns out that $u_x^*$'s can serve as desirable embeddings of $x$'s because they exhibit clustering structure in Euclidean space that resembles the clustering structure of the graph $G(\mathcal{X}, w)$.

### 3.2 From spectral decomposition to spectral contrastive learning

The embeddings $u_x^*$ obtained by eigendecomposition are nonparametric—a $k$-dimensional parameter is needed for every $x$—and therefore cannot be learned with a realistic amount of data. The embedding matrix $F^\star$ cannot be even stored efficiently. Therefore, we will instead parameterize the rows of the eigenvector matrix $F^\star$ as a neural net function, and assume embeddings $u_x^*$ can be represented by $f(x)$ for some $f \in \mathcal{F}$, where $\mathcal{F}$ is the hypothesis class containing neural networks. As we'll show in Section 4, this allows us to leverage the extrapolation power of neural networks and learn the representation on a finite dataset.

Next, we design a proper loss function for the feature extractor $f$, such that minimizing this loss could recover $F^\star$ up to some linear transformation. As we will show in Section 4, the resulting population loss function on $f$ also admits an unbiased estimator with finite training samples. Let $F$ be an embedding matrix with $u_x$ on the $x$-th row, we will first design a loss function of $F$ that can be decomposed into parts about individual rows of $F$.

We employ the following matrix factorization based formulation for eigenvectors. Consider the objective

$$\min_{F \in \mathbb{R}^{N \times k}} \mathcal{L}_{\mathrm{mf}}(F) := \left\| \overline{A} - FF^\top \right\|_F^2. \tag{4}$$

By the classical theory on low-rank approximation (Eckart–Young–Mirsky theorem [19]), any minimizer $\widehat{F}$ of $\mathcal{L}_{\mathrm{mf}}(F)$ contains scaling of the largest eigenvectors of $\overline{A}$ up to a right transformation—for some orthonormal matrix $R \in \mathbb{R}^{k \times k}$, we have $\widehat{F} = F^\star \cdot \mathrm{diag}([\sqrt{\gamma_1}, \ldots, \sqrt{\gamma_k}])Q$. Fortunately, multiplying the embedding matrix by any matrix on the right and any diagonal matrix on the left does not change its linear probe performance, which is formalized by the following lemma.

**Lemma 3.1.** *Consider an embedding matrix $F \in \mathbb{R}^{N \times k}$ and a linear classifier $B \in \mathbb{R}^{k \times r}$. Let $D \in \mathbb{R}^{N \times N}$ be a diagonal matrix with positive diagonal entries and $Q \in \mathbb{R}^{k \times k}$ be an invertible matrix. Then, for any embedding matrix $\widetilde{F} = D \cdot F \cdot Q$, the linear classifier $\tilde{B} = Q^{-1}B$ on $\widetilde{F}$ has the same prediction as $B$ on $F$. As a consequence, we have*

$$\mathcal{E}(F) = \mathcal{E}(\widetilde{F}). \tag{5}$$

*where $\mathcal{E}(F)$ denotes the linear probe performance when the rows of $F$ are used as embeddings.*

The proof can be found in Section C.1.

The main benefit of objective $\mathcal{L}_{\mathrm{mf}}(F)$ is that it's based on the rows of $F$. Recall that vectors $u_x$ are the rows of $F$. Each entry of $FF^\top$ is of the form $u_x^\top x_{x'}$, and thus $\mathcal{L}_{\mathrm{mf}}(F)$ can be decomposed into a sum of $N^2$ terms involving terms $u_x^\top u_{x'}$. Interestingly, if we reparameterize each row $u_x$ by $w_x^{1/2} f(x)$, we obtain a very similar loss function for $f$ that resembles the contrastive learning loss used in practice [12] as shown below in Lemma 3.2. See Figure 1 (right) for an illustration of the relationship between the eigenvector matrix and the representations learned by minimizing this loss.

We formally define the positive and negative pairs to introduce the loss. Let $\bar{x} \sim \mathcal{P}_{\overline{\mathcal{X}}}$ be a random natural data and draw $x \sim \mathcal{A}(\cdot|\bar{x})$ and $x^+ \sim \mathcal{A}(\cdot|\bar{x})$ independently to form a positive pair $(x, x^+)$. Draw $\bar{x}' \sim \mathcal{P}_{\overline{\mathcal{X}}}$ and $x' \sim \mathcal{A}(\cdot|\bar{x}')$ independently with $\bar{x}, x, x^+$. We call $(x, x')$ a negative pair.[3]

---

[2]We index the matrix $A$, $D$ by $(x, x') \in \mathcal{X} \times \mathcal{X}$. Generally we index $N$-dimensional axis by $x \in \mathcal{X}$.

[3]Though $x$ and $x'$ are simply two independent draws, we call them negative pairs following the literature [3].

**Lemma 3.2** (Spectral contrastive loss). *Recall that $u_x$ is the $x$-th row of $F$. Let $u_x = w_x^{1/2} f(x)$ for some function $f$. Then, the loss function $\mathcal{L}_{\mathrm{mf}}(F)$ is equivalent to the following loss function for $f$, called spectral contrastive loss, up to a additive constant:*

$$\mathcal{L}_{\mathrm{mf}}(F) = \mathcal{L}(f) + \mathrm{const}$$

$$\text{where } \mathcal{L}(f) \triangleq -2 \cdot \mathbb{E}_{x,x^+} \left[ f(x)^\top f(x^+) \right] + \mathbb{E}_{x,x'} \left[ \left( f(x)^\top f(x') \right)^2 \right] \tag{6}$$

The proof can be found in Section C.1.

We note that spectral contrastive loss is similar to many popular contrastive losses [12, 38, 40, 50]. For instance, the contrastive loss in SimCLR [12] can be rewritten as (with simple algebraic manipulation)

$$-f(x)^\top f(x^+) + \log \left( \exp \left( f(x)^\top f(x^+) \right) + \sum_{i=1}^n \exp \left( f(x)^\top f(x_i) \right) \right).$$

Here $x$ and $x^+$ are a positive pair and $x_1, \cdots, x_n$ are augmentations of other data. Spectral contrastive loss can be seen as removing $f(x)^\top f(x^+)$ from the second term, and replacing the log sum of exponential terms with the average of the squares of $f(x)^\top f(x_i)$. We will show in Section 6 that our loss has a similar empirical performance as SimCLR without requiring a large batch size.

### 3.3  Theoretical guarantees for spectral contrastive loss on population data

In this section, we introduce the main assumptions on the data and state our main theoretical guarantee for spectral contrastive learning on population data.

To formalize the idea that $G$ cannot be partitioned into too many disconnected sub-graphs, we introduce the notions of *Dirichlet conductance* and *sparsest $m$-partition*, which are standard in spectral graph theory. Dirichlet conductance represents the fraction of edges from $S$ to its complement:

**Definition 3.3** (Dirichlet conductance). *For a graph $G = (\mathcal{X}, w)$ and a subset $S \subseteq \mathcal{X}$, we define the Dirichlet conductance of $S$ as*

$$\phi_G(S) := \frac{\sum_{x \in S, x' \notin S} w_{xx'}}{\sum_{x \in S} w_x}.$$

We note that when $S$ is a singleton, there is $\phi_G(S) = 1$ due to the definition of $w_x$. We introduce the sparsest $m$-partition to represent the number of edges between $m$ disjoint subsets.

**Definition 3.4** (Sparsest $m$-partition). *Let $G = (\mathcal{X}, w)$ be the augmentation graph. For an integer $m \in [2, |\mathcal{X}|]$, we define the sparsest $m$-partition as*

$$\rho_m := \min_{S_1, \cdots, S_m} \max \{ \phi_G(S_1), \ldots, \phi_G(S_m) \}$$

*where $S_1, \cdots, S_m$ are non-empty sets that form a partition of $\mathcal{X}$.*

When $r$ is the number of underlying classes, we might expect $\rho_r \approx 0$ since the augmentations from different classes almost compose a disjoint $r$-way partition of $\mathcal{X}$. However, for $m > r$, we can expect $\rho_m$ to be much larger. For instance, in the extreme case when $m = |\mathcal{X}| = N$, every set $S_i$ is a singleton, which implies that $\rho_N = 1$.

Next, we formalize the assumption that very few edges cross different ground-truth classes. It turns out that it suffices to assume that the labels are recoverable from the augmentations (which is also equivalent to that two examples in different classes can rarely be augmented into the same point).

**Assumption 3.5** (Labels are recoverable from augmentations). *Let $\bar{x} \sim \mathcal{P}_{\overline{\mathcal{X}}}$ and $y(\bar{x})$ be its label. Let the augmentation $x \sim \mathcal{A}(\cdot | \bar{x})$. We assume that there exists a classifier $g$ that can predict $y(\bar{x})$ given $x$ with error at most $\alpha$. That is, $g(x) = y(\bar{x})$ with probability at least $1 - \alpha$.*

We also introduce the following assumption which states that some universal minimizer of the population spectral contrastive loss can be realized by the hypothesis class.

**Assumption 3.6** (Realizability). *Let $\mathcal{F}$ be a hypothesis class containing functions from $\mathcal{X}$ to $\mathbb{R}^k$. We assume that at least one of the global minima of $\mathcal{L}(f)$ belongs to $\mathcal{F}$.*

Our main theorem bound from above the linear probe error of the features learned by minimizing the population spectral contrastive loss.

**Theorem 3.7.** *Assume the representation dimension $k \geq 2r$ and Assumption 3.5 holds for $\alpha > 0$. Let $\mathcal{F}$ be a hypothesis class that satisfies Assumption 3.6 and let $f^*_{\text{pop}} \in \mathcal{F}$ be a minimizer of $\mathcal{L}(f)$. Then, we have*

$$\mathcal{E}(f^*_{\text{pop}}) \leq \widetilde{O}\left(\alpha / \rho^2_{\lfloor k/2 \rfloor}\right).$$

Here we use $\widetilde{O}(\cdot)$ to hide universal constant factors and logarithmic factor in $k$. We note that $\alpha = 0$ when augmentations from different classes are perfectly disconnected in the augmentation graph, in which case the above theorem guarantees the exact recovery of the ground truth. Generally, we expect $\alpha$ to be an extremely small constant independent of $k$, whereas $\rho_{\lfloor k/2 \rfloor}$ increases with $k$ and can be much larger than $\alpha$ when $k$ is reasonably large. For instance, when there are $t$ sub-graphs that have sufficient inner connections, we expect $\rho_{t+1}$ to be on the order of a constant because any $t + 1$ partition needs to break one sub-graph into two pieces and incur a large conductance. We characterize the $\rho_k$'s growth on more concrete distributions in the next subsection.

Previous works on graph partitioning [2, 29, 31] often analyze the so-called rounding algorithms that conduct clustering based on the representations of unlabeled data and do not analyze the performance of linear probe (which has access to labeled data). These results provide guarantees on the approximation ratio—the ratio between the conductance of the obtained partition to the best partition—which may depend on graph size [2] that can be exponentially large in our setting. The approximation ratio guarantee does not lead to a guarantee on the representations' performance on downstream tasks. Our guarantees are on the linear probe accuracy on the downstream tasks and independent of the graph size. We rely on the formulation of the downstream task's labeling function (Assumption 3.5) as well as a novel analysis technique that characterizes the linear structure of the representations. In Section C, we provide the proof of Theorem 3.7 as well as its more generalized version where $k/2$ is relaxed to be any constant fraction of $k$.

### 3.4 Provable instantiation of Theorem 3.7 to mixture of manifold data

In this section, we exemplify Theorem 3.7 on an example where the natural data distribution is a mixture of manifolds, and the augmentation transformation is adding Gaussian noise.

**Example 3.8** (Mixture of manifolds). *Suppose $\mathcal{P}_{\overline{\mathcal{X}}}$ is mixture of $r \leq d$ distributions $P_1, \cdots, P_r$, where each $P_i$ is generated by some $\kappa$-bi-Lipschitz[4] generator $Q: \mathbb{R}^{d'} \to \mathbb{R}^d$ on some latent variable $z \in \mathbb{R}^{d'}$ with $d' \leq d$ which as a mixture of Gaussian distribution:*

$$x \sim P_i \iff x = Q(z), z \sim \mathcal{N}(\mu_i, \frac{1}{d'} \cdot I_{d' \times d'}).$$

*Let the data augmentation of a natural data sample $\bar{x}$ is $\bar{x} + \xi$ where $\xi \sim \mathcal{N}(0, \frac{\sigma^2}{d} \cdot I_{d \times d})$ is isotropic Gaussian noise with $0 < \sigma \lesssim \frac{1}{\sqrt{d}}$. We also assume $\min_{i \neq j} \|\mu_i - \mu_j\|_2 \gtrsim \frac{\kappa \cdot \sqrt{\log d}}{\sqrt{d'}}$.*

Let the ground-truth label be the most likely mixture index $i$ that generates $x$: $y(x) := \arg\max_i P_i(x)$. We note that the intra-class distance in the latent space is on the scale of $\Omega(1)$, which can be much larger than the distance between class means which is assumed to be $\gtrsim \frac{\kappa \cdot \sqrt{\log d}}{\sqrt{d'}}$. Therefore, distance-based clustering algorithms do not apply. We apply Theorem 3.7 and get the following theorem:

**Theorem 3.9.** *When $k \geq 2r + 2$, Example 3.8 satisfies Assumption 3.5 with $\alpha \leq \frac{1}{\text{poly}(d)}$, and has $\rho_{\lfloor k/2 \rfloor} \gtrsim \frac{\sigma}{\kappa \sqrt{d}}$. As a consequence, the error bound is $\mathcal{E}(f^*_{\text{pop}}) \leq \tilde{O}\left(\frac{\kappa^2}{\sigma^2 \cdot \text{poly}(d)}\right)$.*

The theorem above guarantees small error even when $\sigma$ is polynomially small. In this case, the augmentation noise has a much smaller scale than the data (which is at least on the order of $1/\kappa$). This suggests that contrastive learning can non-trivially leverage the structure of the underlying data and learn good representations with relatively weak augmentation. The proof can be found in Section D.

---

[4] A $\kappa$ bi-Lipschitz function satisfies $\frac{1}{\kappa} \|f(x) - f(y)\|_2 \leq \|x - y\|_2 \leq \kappa \|f(x) - f(y)\|_2$.

## 4 Finite-sample generalization bounds

In Section 3, we provide guarantees for spectral contrastive learning on population data. In this section, we show that these guarantees can be naturally extended to the finite-sample regime with standard concentration bounds. In particular, given a training dataset $\{\bar{x}_1, \bar{x}_2, \cdots, \bar{x}_n\}$ with $\bar{x}_i \sim \mathcal{P}_{\overline{\mathcal{X}}}$, we learn a feature extractor by minimizing the following *empirical spectral contrastive loss*:

$$\widehat{\mathcal{L}}_n(f) := -\frac{2}{n} \sum_{i=1}^{n} \mathbb{E}_{\substack{x \sim \mathcal{A}(\cdot|\bar{x}_i) \\ x^+ \sim \mathcal{A}(\cdot|\bar{x}_i)}} \left[f(x)^\top f(x^+)\right] + \frac{1}{n(n-1)} \sum_{i \neq j} \mathbb{E}_{\substack{x \sim \mathcal{A}(\cdot|\bar{x}_i) \\ x' \sim \mathcal{A}(\cdot|\bar{x}_j)}} \left[\left(f(x)^\top f(x')\right)^2\right].$$

It is worth noting that $\widehat{\mathcal{L}}_n(f)$ is an unbiased estimator of the population spectral contrastive loss $\mathcal{L}(f)$. (See Claim E.2 for a proof.) Therefore, we can derive generalization bounds via off-the-shelf concentration inequalities. Let $\mathcal{F}$ be a hypothesis class containing feature extractors from $\mathcal{X}$ to $\mathbb{R}^k$. We extend Rademacher complexity to function classes with high-dimensional outputs and define the Rademacher complexity of $\mathcal{F}$ on $n$ data as $\widehat{\mathcal{R}}_n(\mathcal{F}) := \max_{x_1,\cdots,x_n \in \mathcal{X}} \mathbb{E}_\sigma \left[\sup_{f \in \mathcal{F}, i \in [k]} \frac{1}{n} \left(\sum_{j=1}^{n} \sigma_j f_i(x_j)\right)\right]$,

where $\sigma$ is a uniform random vector in $\{-1, 1\}^n$ and $f_i(z)$ is the $i$-th dimension of $f(z)$.

Recall that $f_{\text{pop}}^* \in \mathcal{F}$ is a minimizer of $\mathcal{L}(f)$. The following theorem with proofs in Section E.1 bounds the population loss of a feature extractor trained with finite data:

**Theorem 4.1.** *For some $\kappa > 0$, assume $\|f(x)\|_\infty \leq \kappa$ for all $f \in \mathcal{F}$ and $x \in \mathcal{X}$. Let $f_{\text{pop}}^* \in \mathcal{F}$ be a minimizer of the population loss $\mathcal{L}(f)$. Given a random dataset of size $n$, let $\hat{f}_{\text{emp}} \in \mathcal{F}$ be a minimizer of empirical loss $\widehat{\mathcal{L}}_n(f)$. Then, with probability at least $1 - \delta$ over the randomness of data, we have*

$$\mathcal{L}(\hat{f}_{\text{emp}}) \leq \mathcal{L}(f_{\text{pop}}^*) + c_1 \cdot \widehat{\mathcal{R}}_{n/2}(\mathcal{F}) + c_2 \cdot \left(\sqrt{\frac{\log 2/\delta}{n}} + \delta\right),$$

*where constants $c_1 \lesssim k^2\kappa^2 + k\kappa$ and $c_2 \lesssim k\kappa^2 + k^2\kappa^4$.*

We can apply Theorem 4.1 to any hypothesis class $\mathcal{F}$ of interest (e.g., deep neural networks) and plug in off-the-shelf Rademacher complexity bounds. For instance, in Section E.2 we give a corollary of Theorem 4.1 when $\mathcal{F}$ contains deep neural networks with ReLU activation.

The theorem above shows that we can achieve near-optimal population loss by minimizing empirical loss up to some small excess loss. The following theorem characterizes how the error propagates to the linear probe performance mildly under some spectral gap conditions.

**Theorem 4.2.** *Assume representation dimension $k \geq 4r + 2$, Assumption 3.5 holds for $\alpha > 0$ and Assumption 3.6 holds. Recall $\gamma_i$ be the $i$-th largest eigenvalue of the normalized adjacency matrix. Then, for any $\epsilon < \gamma_k^2$ and $\hat{f}_{\text{emp}} \in \mathcal{F}$ such that $\mathcal{L}(\hat{f}_{\text{emp}}) < \mathcal{L}(f_{\text{pop}}^*) + \epsilon$, we have:*

$$\mathcal{E}(\hat{f}_{\text{emp}}) \lesssim \frac{\alpha}{\rho_{\lfloor k/2 \rfloor}^2} \cdot \log k + \frac{k\epsilon}{\Delta_\gamma^2},$$

*where $\Delta_\gamma := \gamma_{\lfloor 3k/4 \rfloor} - \gamma_k$ is the eigenvalue gap between the $\lfloor 3k/4 \rfloor$-th and the $k$-th eigenvalue.*

This theorem shows that the error on the downstream task only grows linearly with the error $\epsilon$ during pretraining. We can relax Assumption 3.6 to approximate realizability in the sense that $\mathcal{F}$ contains some sub-optimal feature extractor under the population spectral loss and pay an additional error term in the linear probe error bound. The proof of Theorem 4.2 can be found in Section E.3.

## 5 Guarantee for learning linear probe with labeled data

In this section, we provide theoretical guarantees for learning a linear probe with *labeled* data. Theorem 3.7 guarantees the existence of a linear probe that achieves a small downstream classification error. However, a priori it is unclear how large the margin of the linear classifier can be, so it is hard to apply margin theory to provide generalization bounds for 0-1 loss. We could in principle control the margin of the linear head, but using capped quadratic loss turns out to suffice and mathematically more

convenient. We learn a linear head with the following *capped quadratic loss*: given a tuple $(z, y(\bar{x}))$ where $z \in \mathbb{R}^k$ is a representation of augmented data $x \sim \mathcal{A}(\cdot|\bar{x})$ and $y(\bar{x}) \in [r]$ is the label of $\bar{x}$, for a linear probe $B \in \mathbb{R}^{k \times r}$ we define loss $\ell((z, y(\bar{x})), B) := \sum_{i=1}^{r} \min\left\{ \left(B^\top z - \vec{y}(\bar{x})\right)_i^2, 1 \right\}$, where $\vec{y}(\bar{x})$ is the one-hot embedding of $y(\bar{x})$ as a $r$-dimensional vector (1 on the $y(\bar{x})$-th dimension, 0 on other dimensions). This is a standard modification of quadratic loss in statistical learning theory that ensures the boundedness of the loss for the ease of analysis [36].

The following Theorem 5.1 provides a generalization guarantee for the linear classifier that minimizes capped quadratic loss on a labeled dataset. The key challenge of the proof is showing the existence of a small-norm linear head $B$ that gives small population quadratic loss, which is not obvious from Theorem 3.7 where only small 0-1 error is guaranteed. Recall $\gamma_i$ is the $i$-th largest eigenvalue of the the normalized adjacency matrix. Given a labeled dataset $\{(\bar{x}_i, y(\bar{x}_i))\}_{i=1}^{n}$ where $\bar{x}_i \sim \mathcal{P}_{\overline{\mathcal{X}}}$ and $y(\bar{x}_i)$ is its label, we sample $x_i \sim \mathcal{A}(\cdot|\bar{x}_i)$ for $i \in [n]$.

**Theorem 5.1.** *In the setting of Theorem 3.7, assume $\gamma_k \geq C_\lambda$ for some $C_\lambda > 0$. Learn a linear probe $\widehat{B} \in \arg\min_{\|B\|_F \leq 1/C_\lambda} \sum_{i=1}^{n} \ell((f_{\text{pop}}^*(x_i), y(\bar{x}_i)), B)$ by minimizing the capped quadratic loss subject to a norm constraint. Then, with probability at least $1 - \delta$ over random data, we have*

$$\Pr_{\bar{x} \sim \mathcal{P}_{\overline{\mathcal{X}}}} \left( \bar{g}_{f_{\text{pop}}^*, \widehat{B}}(\bar{x}) \neq y(\bar{x}) \right) \lesssim \frac{\alpha}{\rho_{\lfloor k/2 \rfloor}^2} \cdot \log k + \frac{r}{C_\lambda} \cdot \sqrt{\frac{k}{n}} + \sqrt{\frac{\log 1/\delta}{n}}.$$

Here the first term is the population error from Theorem 3.7. The last two terms are the generalization gap from standard concentration inequalities for linear classification and are small when the number of labeled data $n$ is polynomial in the feature dimension $k$. We note that this result reveals a trade-off when choosing the feature dimension $k$: when $n$ is fixed, a larger $k$ decreases the population contrastive loss while increases the generalization gap for downstream linear classification. The proof of Theorem 5.1 is in Section F.

# 6    Experiments

We test spectral contrastive learning on benchmark vision datasets. We minimize the empirical spectral contrastive loss with an encoder network $f$ and sample fresh augmentation in each iteration. The pseudo-code for the algorithm and more implementation details can be found in Section A.

**Encoder / feature extractor.** The encoder $f$ contains three components: a backbone network, a projection MLP and a projection function. The backbone network is a standard ResNet architecture. The projection MLP is a fully connected network with BN applied to each layer, and ReLU activation applied to each except for the last layer. The projection function takes a vector and projects it to a sphere ball with radius $\sqrt{\mu}$, where $\mu > 0$ is a hyperparameter that we tune in experiments. We find that using a projection MLP and a projection function improves the performance.

**Linear evaluation protocol.** Given the pre-trained encoder network, we follow the standard linear evaluation protocol [14] and train a supervised linear classifier on frozen representations, which are from the ResNet's global average pooling layer.

**Results.** We report the accuracy on CIFAR-10/100 [26] and Tiny-ImageNet [27] in Table 1. Our empirical results show that spectral contrastive learning achieves better performance than two popular baseline algorithms SimCLR [12] and SimSiam [14]. In Table 2 we report results on ImageNet [18] dataset, and show that our algorithm achieves similar performance as other state-of-the-art methods. We note that our algorithm is much more principled than previous methods and doesn't rely on large batch sizes (SimCLR [12]), momentum encoders (BYOL [21] and MoCo [22]) or additional tricks such as stop-gradient (SimSiam [14]).

# 7    Conclusion

In this paper, we present a novel theoretical framework of self-supervised learning and provide provable guarantees for the learned representations on downstream linear classification. We hope our study can facilitate future theoretical analyses of self-supervised learning and inspire new practical algorithms. For instance, one interesting future direction is to test the topology of the augmentation graph on empirical data distributions and design algorithms using tools from graph theory.

| Datasets | CIFAR-10 | | | CIFAR-100 | | | Tiny-ImageNet | | |
|---|---|---|---|---|---|---|---|---|---|
| Epochs | 200 | 400 | 800 | 200 | 400 | 800 | 200 | 400 | 800 |
| SimCLR (repro.) | 83.73 | 87.72 | 90.60 | 54.74 | 61.05 | 63.88 | **43.30** | **46.46** | 48.12 |
| SimSiam (repro.) | 87.54 | **90.31** | 91.40 | 61.56 | 64.96 | 65.87 | 34.82 | 39.46 | 46.76 |
| Ours | **88.66** | 90.17 | **92.07** | **62.45** | **65.82** | **66.18** | 41.30 | 45.36 | **49.86** |

Table 1: Top-1 accuracy under linear evaluation protocal.

| | SimCLR | BYOL | MoCo v2 | SimSiam | Ours |
|---|---|---|---|---|---|
| acc. (%) | 66.5 | 66.5 | 67.4 | 68.1 | 66.97 |

Table 2: ImageNet linear evaluation accuracy with 100-epoch pre-training. All results but ours are reported from [14]. We use batch size $384$ during pre-training.

## Acknowledgements

We thank Margalit Glasgow, Ananya Kumar, Jason D. Lee, Sang Michael Xie, and Guodong Zhang for helpful discussions. CW acknowledges support from an NSF Graduate Research Fellowship. TM acknowledges support of Google Faculty Award and NSF IIS 2045685. We also acknowledge the support of HAI and the Google Cloud. Toyota Research Institute ("TRI") provided funds to assist the authors with their research but this article solely reflects the opinions and conclusions of its authors and not TRI or any other Toyota entity.

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
