# OpenReview forum: "Provable Guarantees for Self-Supervised Deep Learning with Spectral Contrastive Loss"
_NeurIPS.cc/2021/Conference — NeurIPS 2021 Oral_

### Official Review · Reviewer_HpLj · 2021-07-12

**Rating:** 7
**Confidence:** 3

**Summary:**

This paper theoretically analyzes contrastive learning relying on a concept of population augmentation graph. In the graph, two augmented samples are connected if they can be augmented from the same original instance. Based on this setup,
- This paper shows that spectral decomposition can be reformulated as a contrastive learning objective. Furthermore, this paper shows that minimizing the objective can guarantee a high linear probe accuracy under some mild assumptions.
- This paper shows the guarantee can be extended to the finite-sample regime and learning the linear probe with labeled data.
- This paper demonstrates the efficacy of the proposed spectral contrastive objective empirically in the standard setup: the proposed objective achieves similar performance as other state-of-the-art SSL methods.


**Limitations And Societal Impact:**

Yes. This paper addressed the limitations and potential negative societal impact.


**Main Review:**

Strengths
- This paper is well-written and well-organized. I believe that this paper has good readability for not only theorists but also practitioners.
- The proposed concept of the population augmentation graph and the spectral decomposition approach is convincing.
- It seems that Assumption 3.5 is practically reasonable for contrastive learning analysis. This is because SSL methods often assume that two augmented samples have the same semantic (i.e., underlying class) information; in other words, they also assume that correct predictions from the augmented samples are available (with high probability).
- This paper extends the guarantee (Theorem 3.7) to more practical scenarios. I feel that the result in Section 5 is interesting because the setting is almost the same as the standard linear evaluation benchmark.
- The proposed spectral contrastive loss shows competitive performance compared to state-of-the-art SSL methods. I think more empirical studies could improve the performance.

Concerns & Questions
- There is no empirical study related to the theoretical results.
  - This paper considers $k$ as a fixed constant, but one can choose $k$ arbitrarily in practice. In this case, how can we choose $k$? In Theorem 3.7 and 5.1, there is a trade-off in the upper bound. Therefore, we may need to choose proper $k$ for achieving the best performance. Can we observe the trade-off empirically, as stated in the theorems?
  - I'm also interested in the $\rho_k$'s growth because $\rho_k$ is the main factor of the error bound. Although L246 said that the growth would be characterized in Section 3.4, I cannot fully understand which point explains the growth. Could you provide more details? Also, can we approximate $\rho_k$ empirically? I wonder whether the error bound is still meaningful in practice or not.
- What is the role of the projection MLP in a theoretical perspective? The projection MLP is commonly used in self-supervised learning, and this work also uses it in the provided experiments (Section 6). However, the guarantees are obtained without considering the projection MLP.
- Can we omit $\log k$ in Theorem 3.7? By definition, $\rho_k\le 1$. Also, intuitively, $\rho_k\approx 1$ when $k$ is sufficiently large. In this case, the upper bound becomes large due to $\log k$. In my opinion, it would be better to do not hide $\log k$ in Theorem 3.7 and do not use the $\tilde{O}$ notation. This might confuse some readers: for example, one can think the upper bound goes to $\alpha$ for sufficiently large $k$.

Typos
- In Eq. (7), the second term might be $f(x)^\top f(x')$.
- In L215, Why $\phi_G(\{x\})=1$? In my opinion, $\phi_G(\{x\})=\frac{\sum_{x'\neq x} w_{xx'}}{\sum_{x'} w_{xx'}}<1$ because $w_{xx}>0$ for any $x\in\mathcal{X}$.
- In Eq. (11), does the second equality hold? I think the $\sqrt{w_x}$ terms should be in the numerator.

== after rebuttal ==

I sincerely appreciate your efforts to address my concerns. Like other reviewers, I think this paper's theoretical contribution is convincing, and I like the empirical results based on the theoretical findings. Hence, I would like to keep my positive score.


**Time Spent Reviewing:**

8

---

> ### Author Response · Authors · 2021-08-10
> **Response**
>
> We thank the reviewers for their comments, and for noting that "the proposed concept of the population augmentation graph and the spectral decomposition approach is convincing" and that "Assumption 3.5 is practically reasonable". Answers to specific points are below:
>
> >*"we may need to choose proper k for achieving the best performance. Can we observe the trade-off empirically, as stated in the theorems? "*
>
> Yes, empirically we also observe that the optimal downstream classification performance is achieved with some proper choice of k, while setting k to be too large or too small degrades the downstream classification accuracy.
>
> >*"I'm also interested in the rho_k's growth because rho_k is the main factor of the error bound. Although L246 said that the growth would be characterized in Section 3.4, I cannot fully understand which point explains the growth. ... Also, can we approximate rho_k empirically?"*
>
> We agree with the reviewer that rho_k is the main factor of the error bound, and our error bound is only small if $\rho_k$ is relatively large (which happens when there are no more than k tightly-connected subgraphs). To characterize the growth of $\rho_k$ (i.e., showing that $\rho_k$ is large when $k$ is small), we instantiate our theory on a mixture-of-manifolds example in Section 3.4, where each class corresponds to a d-dimensional manifold generated by a continuous transformation of a Gaussian distribution, and the data augmentation is small Gaussian perturbation with scale $\sigma$. When the manifolds are nicely separated (at least distance $1/\sqrt{d}$ between each other), we give an lower bound $\rho_{\lfloor k/2\rfloor}\ge\frac{\sigma}{\sqrt{d}}$ (Thm 3.9) for any $k\ge 2r+2$ ($r$ is the number of ground truth classes). By plugging this lower bound into Theorem 3.7, we show that our error bound is indeed small for the mixture-of-manifolds example (Thm 3.9).
>
> We agree with the reviewer that empirically estimating $\rho_k$ can help us better understand how meaningful the bound is. However, it is challenging because estimating $\rho_k$ requires knowing the population data manifolds. One possibility is to leverage GANs as approximations of the population data (see [1] for some preliminary investigations using this technique). We believe that this is an important open question which we’ll leave for future work.
>
> >*"What is the role of the projection MLP in a theoretical perspective?"*
>
> We believe that understanding the role of projection MLP is an important open question in self-supervised learning. Unfortunately, our current analysis doesn’t explain its empirical success. That being said, we observe a strong correlation between the downstream accuracy between the features trained with and without projection MLP, using the same loss objective. Therefore, we believe that our analysis still sheds light on the empirical success of self-supervised learning methods even though there’s no projection MLP involved in the theory.
>
> >*"Why phi_G(x)=1?"*
>
> We thank the reviewer for pointing out this typo. We agree with the reviewer that $\phi_G(x)$ can be slightly smaller than $1$ when $w_{xx}>0$ for some $x$. One might still intuitively think of $\phi_G(x)\approx 1$ because the probability of two augmentations being the same is extremely small. We will update this in the next version of the paper.
>
> >*"In Eq. (11), does the second equality hold? I think the $\sqrt{w_x}$ terms should be in the numerator."*
>
> Yes, we thank the reviewer for pointing out this typo! We will correct this equation in the next revision.
>
>
> [1] Wei, Colin, et al. "Theoretical analysis of self-training with deep networks on unlabeled data." arXiv preprint arXiv:2010.03622 (2020).

---

### Official Review · Reviewer_N3Yd · 2021-07-13

**Rating:** 8
**Confidence:** 4

**Summary:**

The paper studies the theoretical foundations of contrastive self-supervised learning. To do so, the authors study contrastive learning through the lens of the augmentation graph - the graph of augmented examples, with edges connecting "similar" examples. The authors suggest a novel contrastive loss function, s.t. minimizing this loss is equivalent to spectral clustering on the augmentation graph. Using this, the authors prove that if the augmentation graph has few edges that cross between sub-graphs of different labels, then using a linear readout on the learned representation, a small loss can be achieved w.r.t the supervised objective. The authors also analyze the sample complexity of learning the representation and learning the linear probe from sampled data. Finally, the authors show empirically that the suggested contrastive loss performs similarly to some baseline contrastive loss functions.

**Limitations And Societal Impact:**

Yes

**Main Review:**

Overall, I think this is a very good paper. It is very clearly written, the results are novel, and give a new characterization of contrastive learning using the augmentation graph. The connection between self-supervised learning, graph theory and spectral clustering is very interesting, and to my knowledge this is the first work that connects these fields. The theoretical results are very clean, and their presentation makes it easy for the reader to understand and appreciate the results.

One thing that I think is still missing is a sample complexity analysis of learning both the representation and the linear probe. The paper studies the sample complexity of learning the representation, and then using the optimal linear classifier, or otherwise learning the linear probe from a finite sample using the optimal representation. However, in practice self-supervised learning uses large unlabeled data to learn the representation, and then a small labeled dataset to learn the linear classifier. Since the sample complexity is analyzed separately for the representation and the linear probe, it is not clear whether or not we gained anything from using self-supervised learning over supervised learning. That is - we have a trivial sample complexity bound for supervised learning of Bf(x), e.g. using Radamacher complexity. Is this sample complexity larger than the sample complexity of learning only B from labeled data? If so, what is the cost in unlabeled data?

Minor:
Line 189 - x_{x'} should be u_{x'}

**Time Spent Reviewing:**

3

---

> ### Author Response · Authors · 2021-08-10
> **Response**
>
> We thank the reviewers for their comments, and for noting that "the connection between self-supervised learning, graph theory, and spectral clustering is very interesting". Our response to the reviewer’s question about sample complexity is below:
>
> >*"One thing that I think is still missing is a sample complexity analysis of learning both the representation and the linear probe … whether or not we gained anything from using self-supervised learning over supervised learning. "*
>
> We thank the reviewer for suggesting a result about training both the representation and linear probe with finite samples. To achieve this, we can simply combine our Theorem 4.2 (which is for contrastive learning with finite unlabeled data) and Theorem 5.1 (which is linear probing with finite labeled data). The final errors will essentially consist of the sum of the errors from the two Theorems, that is, $\alpha/\rho^2_{\lfloor k/2 \rfloor} + \frac{k}{\Delta_\gamma^2} \cdot \hat{R}_{n_u}(\mathcal{F}) + \frac{r}{C_\lambda}\cdot \sqrt{\frac{k}{n_l}}$, where $n_u$ and $n_l$ are the number of unlabeled and labeled data, respectively. Here the first term comes from Theorem 3.7 which is about the intrinsic property of the data, the second term combines the generalization error of the unsupervised loss (Theorem 4.1) and the optimization error of the unsupervised loss (Theorem 4.2), and the third one is the generalization error of the linear probe. We will add this extension in the next revision.
>
> Indeed, this bound suggests that self-supervised learning achieves a better sample complexity bound (in terms of labeled data used) than supervised learning. Intuitively, this is because we only use labeled data for the linear probe which has a small number of parameters, and the unsupervised learning generalization error depends on the number of *unlabeled* data.
>
> More concretely, ignoring the first term in the bound above (which don’t depend on the number of data) and treating $r$ (number of classes), $\Delta_\gamma$ (defined in Thm 4.2) and $C_\lambda$ (defined in Thm. 5.1) as constants, the final error is approximately $O(\sqrt{C/n_u}+\sqrt{k/n_l})$, where $C$ is some complexity measure of the neural network for the feature extractor (e.g., assuming $\hat{R}_{n_u}(\mathcal{F}) = \sqrt{C/n_u}$). On the other hand, directly training the neural network on labeled data gives an error bound $O(\sqrt{C/n_l})$. So to achieve $\epsilon$ error, self-supervised learning requires $k/\epsilon^2$ labeled data and $C/\epsilon^2$ unlabeled data, whereas supervised learning requires $C/\epsilon^2$ labeled data. When the model complexity $C$ is much larger than the representation dimension k (which is typically the case for neural networks), self-supervised learning requires much fewer labeled data. Furthermore, the required number of unlabeled data is at most constant factor more than the number of labeled data required in the supervised training paradigm.
>
> >*"Line 189 - x_{x'} should be u_{x'}"*
>
> Yes, we thank the reviewer for pointing this out. We’ll address this typo in the next revision.

---

### Official Review · Reviewer_7GCn · 2021-07-14

**Rating:** 8
**Confidence:** 4

**Summary:**

This paper derives generalization bounds for learning using a contrastive objective under realistic assumptions on the positive pair generation process. The approach taken is to consider the so-called augmentation graph - the set of nodes comprises all augmented views of population distribution, with edges connecting nodes that correspond to different views of the same input datapoint. The core assumption made is that this graph cannot be split into a large number of disconnected subgraphs. This set-up aligns well with the intuition that in order to generalize, the contrastive notion of “similarity” must extent beyond the purely single-instance-level, and must somehow connect distinct inputs points.

My view is that this work is the most significant advance in the statistical learning theory of contrastive learning since the paper “A theoretical analysis of contrastive unsupervised representation learning” by Arora et al. This paper under review addresses the primary weakness in the work by Arora et al., which used an unrealistic assumption on positive pair generation. I will strongly advocate for acceptance of this paper.



**Limitations And Societal Impact:**

In my view the authors have adequately addressed the limitations and potential negative impacts of their work.

**Main Review:**


## Discussion:

Prior work on contrastive and/or self-supervised learning theory typically either 1) assumes positive pairs are near-independent when conditioned on class label, or 2) is based on an information theoretic analysis (this is to the best of my knowledge, at least). The first case is clearly unrealistic, and the second suffers from at least a couple of drawbacks - a) the degree to which the success of contrastive learning (particular InfoNCE) can be attributed to its relation to mutual information has been experimentally disputed [1], and b) do not adequately explain why linear probes are sufficient to yield effective downstream performance.

This paper avoids both of these drawbacks by proposing a new set of assumptions on positive pair generation (which in my view are much more realistic that previous assumption used) and directly analyzing a contrastive loss (which they actually propose in this work) instead of relating contrastive leanring to information theoretic concepts.

**The assumptions on positive pair generation:**

Contrastive learning algorithms work on the level of individual instances - enforcing the similarity of embeddings of different views of *the same* individual instance. Despite this, the performance under linear probe is high on many downstream tasks. This empirical fact strongly indicates that learned encoders must be doing more than learning instance-level invariances. So how does the instance-level approach of contrastive methods extend to enforce relations between the embeddings of distance inputs?

The paper under review takes a step towards answering this question by showing that under what I consider to be a fairly intuitive set of assumptions (but which are hard to verify empirically) the error of a linear probe can be bounded. The plausibility of these assumptions is the key to this work. If other reviewers are critical, it will almost certainly be due to some misgivings around these assumptions.

As well as a fresh perspective on the assumptions underlying positive pair generation, this paper imports new tools (from spectral theory) to the analysis of contrastive leanring. The introduction of “new” analysis tools (new in the context of contrastive learning) is welcome in my eyes, and my be of use to others working on the theory of contrastive learning.

I should note that due to time pressure I have not carefully checked the soundness of the proofs. From what I can tell so far I have seen nothing out of order.

---
## Minor criticisms:

I have already made my overall assessment of this work clear. This section contains minor criticisms.

- The finite sample bounds in Sec 4 are still somewhat separated from the objective used in practice. The finite sample objective in Sec 4 still takes expectations over the augmentation distribution: in practice this is approximated by drawing a single sample.
- A very minor gripe: there are some suboptimalities in the experimental protocols. E.g. the authors report for SimCLR an accuracy of 46.5% on tinyImageNet with ResNet-50 backbone and 400 epochs of training, and batch size 512. In my own experiments in the past I have obtained over 53% using the same backbone, epochs, and batch size. This isn’t a big deal, though, and the experiments as they are suffice to illustrate the important point - that the proposed spectral loss can learn encoders with comparable performance to other popular methods of the day.

---
## Clarity:

- l.179 I think R=Q here?
- l.48 I suppose \cal Z^+ denotes the positive integers. I would suggest using the more standard \mathbb{Z}^+, or simply saying “positive integer m” instead.

---
## Questions:
- You analysis, like many others, focuses on downstream classification tasks. How might your viewpoint relate to non-classifcation tasks that contrastive methods also enjoy empirically success on - e.g. object detection and image segmentation?


**Time Spent Reviewing:**

3hrs

---

> ### Author Response · Authors · 2021-08-10
> **Response**
>
> We thank the reviewers for their comments, and will incorporate the feedback and address minor issues in the revision. The reviewer notes that "this work is the most significant advance in the statistical learning theory of contrastive learning since the paper 'A theoretical …' by Arora et al." and "this paper addresses the primary weakness in the work by Arora et al." Answers to specific points are below:
>
> >*“The finite sample objective in Sec 4 still takes expectations over the augmentation distribution, but in practice this is approximated by drawing a single sample.”*
>
> We thank the reviewer for pointing out this potential difference between our objective and that used in reality. Actually, our choice to define the objective with an expectation over augmentation distribution is purely for the purpose of cleaner exposition. Our analysis directly applies to the case when only a finite number of augmentations for each natural data are involved in the definition of loss. For instance, in the Appendix, we introduce an objective where only one augmentation is sampled for each natural data (Definition E.3), and show that achieving small empirical loss on this objective leads to a similar error bound as that in Theorem 4.1 (Equation (19) in the Appendix).
>
> >*“You analysis, like many others, focuses on downstream classification tasks. How might your viewpoint relate to non-classification tasks ... e.g. object detection and image segmentation?”*
>
> We thank the reviewer for suggesting studying other non-classification tasks theoretically, which we believe is an important open question for future research. Currently, there is limited theoretical work on segmentation and object detection, perhaps partly because it's challenging to formulate these tasks with proper mathematical assumptions. Empirically, researchers have observed that representations with better classification performance usually also have better performances on other tasks like object detection and image segmentation [1]. We hope that our theory about classification can also help shed light on understanding those other downstream tasks in the future.
>
> >*“l.179 I think R=Q here?”*
>
> Yes, we thank the reviewer for pointing this out. We’ll address this typo in the next revision.
>
>
> [1] Chen, Xinlei, and Kaiming He. "Exploring simple siamese representation learning." Proceedings of the IEEE/CVF Conference on Computer Vision and Pattern Recognition. 2021.

---

### Official Review · Reviewer_1aqU · 2021-07-20

**Rating:** 7
**Confidence:** 3

**Summary:**

The paper studies how does contrastive learning works from a theoretical perspective focusing on the spectral properties of the graph of the data induced by augmentations. They propose a new loss that is close up to a constant to the matrix factorization of the normalized adjacency matrix of the augmentations graph, under some assumptions they show that minimizing this loss provides downstream classification guarantees. Then, using standard Rademacher analysis they move from population guarantees to finite sample guarantees. Finally, they show empirically that their loss gives results close to state of the art.

**Limitations And Societal Impact:**

yes

**Main Review:**

The paper is interesting and original. Connecting spectral graph theory with SSL is a surprising finding that I would not have foreseen a-priori. Moreover, the theory seems sound to my understanding and the paper is well written and accessible. I have few minor reservations:

The authors assume that "G cannot be partitioned into too many disconnected sub-graphs". This assumption seems rather strong for real data. Every *real* image therefore will be a disconnected subgraph with its augmentations, especially so for high resolution images. Where does this come about in the bounds? As far as I understand this should appear in \rho in Theorem 3.7. However, for example, if \alpha is 0 the bound should be 0. This seems unreasonable to me. What am I missing here?

A related issue that I'm wondering about is when does this construction break? Do assumptions 3.5 and 3.6 suffice to guarantee good downstream performance (assuming perhaps that the Rademacher complexity is small)? If so why don't we get 99% accuracy on ImageNet in the experiments?

Other comments:
* Line 179 I think there is some confusion between R and Q.
* Line 189 Is this u_x'?

======

I would like to thank the authors for their detailed an enlightening comments that addressed my main concerns. After reading the comments and discussion with the other reviewers I would like to raise my score to 7.

**Time Spent Reviewing:**

6 hours

---

> ### Author Response · Authors · 2021-08-10
> **Response**
>
> We thank the reviewers for their comments and will incorporate the feedback and address minor issues in the revision. The reviewer notes that this "paper is interesting and original", and finds that "connecting spectral graph theory with SSL is a surprising finding" that the reviewer "would not have foreseen a-priori". Answers to specific points are below:
>
> >*"The authors assume that 'G cannot be partitioned into too many disconnected sub-graphs'. This assumption seems rather strong for real data." "Every real image will be a disconnected subgraph with its augmentations, especially so for high resolution images."*
>
> We’d like to clarify that this appears to be a misunderstanding of our assumption. Though we agree with the reviewer if the graph is built with empirical samples, we clarify that our assumptions are made on the *population* augmentation graph. Making connectivity assumptions on the population graph (instead of the empirical graph) is an important innovation of our paper, because two data points in the same class are very likely to be connected via a sequence of (unseen) data points in the population, but they cannot be connected via a sequence of training examples (because training examples are generally far away from each other.)
>
> A concrete example is shown in Figure 1 (left). The figure shows five augmentations of dogs of the Brittany breed (denoted by $C_1$, …, $C_5$) that form a connected chain. Even though $C_1$ and $C_{5}$ look quite different, each consecutive pair $(C_i, C_{i+1})$ is similar and shares a common original image. Note that the intermediate images $C_2, C_3, C_4$ are not necessarily in the training data set---they only have to show up in the support of the population distribution. In fact, for any two Brittanies $A$ and $B$, we believe there exists $C_2,\dots, C_{m-1}$ in the population distribution’s support such that $A,C_2,\dots,C_{m-1}, B$ forms a chain where consecutive pairs are connected. In other words, we can transform one dog into the other by gradually changing the position, shape, coat color, and patterns, etc.
>
> As the connectivity of the augmentation graph appears to be a major concern of the reviewer, we respectfully ask the reviewer to consider increasing the score if they are satisfied with the explanations above.
>
>
> >*"if \alpha is 0 the bound should be 0"*
>
> Yes, indeed, in the extreme case when $\alpha=0$, our bound in Theorem 3.7 gives 0 error so long as $\rho_{\lfloor k/2 \rfloor}>0$. Note that $k/2$ needs to be bigger than the number of disconnected subgraphs so that $\rho_{\lfloor k/2 \rfloor}>0$. Achieving 0 error is reasonable because, in Theorem 3.7, we are operating under the situation that the population contrastive loss is optimized perfectly. (Theorem 4.1 and Theorem 4.2 deal with the additional error introduced by the empirical contrastive loss, in which case the error may not be 0 even when $\alpha=0$.) Our next response is also related to this question.
>
>
> >*"when does this construction break? Do assumptions 3.5 and 3.6 suffice to guarantee good downstream performance (assuming perhaps that the Rademacher complexity is small)? why don't we get 99% accuracy on ImageNet in the experiments?"*
>
> Yes, Assumptions 3.5 and 3.6 suffice for guaranteeing the downstream performance, assuming that the Rademacher complexity is small (or equivalently the number of unlabeled pretraining data is large), and the number of training data in the downstream linear probe is larger than k. We note that Theorem 3.7 is the key theorem but it’s in the idealistic setting of infinite unlabeled pretraining data. In the finite data setting, there are indeed more errors caused in Theorem 4.1 & 4.2 (where the errors stem from finite unlabeled data and optimization error) and Theorem 5.1 (where the error stems from finite linear probe data). We believe that these errors are responsible for the empirical imperfection on ImageNet. Another possibility is that the realizability assumption (Assumption 3.6) may not be exactly achieved by current practical architectures, because the practical models have limited capacity (and a more powerful net may cause more generalization error). We believe that investigating how each of these possibilities contributes to the empirical error is an important future direction.
>
>
> >*"Line 179 I think there is some confusion between R and Q."*
>
> We thank the reviewer for pointing out this typo. The notation Q should have been R in this sentence, we will address this in the next revision.
>
>
> >*---"Line 189 Is this u_x'?"*
>
> Yes, we thank the reviewer for pointing this out. We’ll address this typo in the next revision.

---

> > ### Comment · Reviewer_1aqU · 2021-08-12
> > **Response**
> >
> > I would like to thank the authors for their response and for clarifying some of the questions.
> >
> > I'm still unconvinced by the 'all dogs of the same breed will be in the same connected component' (up to some sparse connection to other components) even when we talk about the population graph. In the example the authors provided (fig 1), the augmentations belong only to one parent image (and not two as mentioned above). I agree that if we make the augmentations strong enough we will reach connectivity but will it be sparse wrt other components? That is, if every brittany dog is connected, why are they not connected to all cats as well? It seems that there should be a delicate balance and in my opinion the response above does not address it.

---

> > > ### Author Response · Authors · 2021-08-12
> > > **Response**
> > >
> > > We thank the reviewer for their comments and additional questions.
> > >
> > > We would like to clarify that the augmentations in Fig. 1 do belong to *different* parent images. There are four different natural images in Fig. 1, which generate the five augmented images. This happens partly because our augmentation in Fig. 1 involves random cropping, and there exists similar patches between 2 different Brittany dog images.
> > >
> > > In addition, we would like to clarify two points in the following paragraphs:
> > > * The connectivity within Brittany dogs and separations from cats do *not* necessarily have to rely on the random cropping or strong augmentations. (Though stronger augmentations do help.)
> > > * Perhaps the reviewer was also suggesting that the natural images in Fig. 1, though different, are the images of the same dog. This is true in our Figure (because the authors only have one dog to film); but the connectivity will apply to different images of different Brittany dogs as well, as argued below.
> > >
> > > To justify these two points, let’s consider perhaps the weakest possible augmentations: adding perturbation with a *very small* $\ell_2$ norm. In other words, two images are only connected when their $\ell_2$ distance is very small. Let’s also consider two images of two different Brittany dogs, A and B. The two dogs perhaps differ by size, positions, coat pattern, color, appearances, facial expression, etc. We argue that it’s still possible to transform A via a sequence of small deformations into the image B, such that the intermediate images $C_2,... C_{m-1}$ are meaningful images of Brittany dogs that can show up in the population data and $C_{i}, C_{i+1}$ are close. However, if one aims to connect a Brittany dog with a cat via a sequence of images, there must be a big jump at some places, or some of the intermediate images are neither cats or dogs and thus cannot show up in the support of the population distribution.
> > >
> > > To visualize this, please see the [link](https://www.dropbox.com/sh/k2dbd1a2kmm03dl/AAAzJoXYXPGknjJW5qqhogCKa?dl=0) for how to connect two quite different Yorkshire terrier dog images with a sequence of Yorkshire terrier images, such that any consecutive pair involves only a small perturbation. (These images were originally generated by other researchers (whom the authors don’t know) using GANs and shared on YouTube for other purposes. We are only using them to demonstrate the connectivity. We have to use GAN generated images (which are not perfect) because we don’t have access to the population distribution.)
> > >
> > > We will incorporate these discussions in the revision as well.

---

> > > ### Author Response · Authors · 2021-08-27
> > > **Response**
> > >
> > > We would like to thank the reviewer again for the comments and questions. We hope that our latest response above has addressed the questions. We would also be more than happy to respond in more detail in case there are still remaining concerns!

---

### Decision · Program_Chairs · 2021-09-27

**Decision:**

Accept (Oral)

**Comment:**

This paper provides a new theoretical framework for contrastive learning. In particular, the authors find a new type of assumption, connecting spectral graph theory with self-supervised contrastive learning, which is much more realistic than what is common in prior statistical theory for contrastive learning. All reviewers agree on the significance of theoretical contribution provided by the paper. One of major concerns was on the connectivity assumption of the augmentation graph raised by Reviewer 1aqU. As the authors did in their rebuttal, it is quite useful to provide some concrete visual examples in the final draft, to avoid some misunderstanding. AC thinks it is also useful to provide some (real-world or synthetic) scenarios when the assumption breaks, possibly with supporting experiments (to see how much the assumption is crucial). In overall, AC thinks that the paper is very well written and could be a pioneering work, not only for theoretical purposes, but also for some better algorithmic solutions in the future.